# Lexical Borrowings from Spanish into Wayuunaiki: Contact, Classification, and Motivations

**Johan De La Rosa Yacomelo** [1,*], **Rudecindo Ramírez González** [2], **Leonel Viloria Rodríguez** [3],
**Wendy Valdez Jimenez** [4] and **Jesús Guerra-Lyons** [4]

[1] Department of Foreign Languages, Universidad del Norte, Barranquilla 081007, Colombia
[2] Faculty of Education, Universidad de la Guajira, Riohacha 440003, Colombia; rramirez@uniguajira.edu.co
[3] Department of Social Sciences and Humanities, Corporación Universitaria del Caribe,
   Sincelejo 700001, Colombia; leonel.viloriar@cecar.edu.co
[4] Department of Spanish, Universidad del Norte, Barranquilla 081007, Colombia;
   valdezw@uninorte.edu.co (W.V.J.); jdguerra@uninorte.edu.co (J.G.-L.)
[*] Correspondence: jyacomelo@uninorte.edu.co; Tel.: +57-53509794 (ext. 4041)

**Abstract:** This study identifies and analyzes lexical borrowings from Spanish into Wayuunaiki (an Arawak language spoken in Colombia and Venezuela). The analysis, based on bibliographic documentation and fieldwork, focuses on the borrowability and semantic domains of Spanish loanwords and the factors motivating Spanish loanword adoption into Wayuunaiki. The results show that, despite the typological distance, Wayuunaiki is prone to adopting lexical items from Spanish, as evident in the moderate number of lexical borrowings identified. A key motivating factor for Spanish loanword incorporation is Wayuunaiki speakers' need for lexical items to refer to new concepts adopted from the dominant Spanish-speaking culture. This phenomenon is partly due to the contact dynamics between Wayuunaiki and Spanish. Besides expanding on the description of the Wayuunaiki language and its contact with Spanish, this study contributes to identifying factors and motivations favoring the adoption of loanwords between typologically distant languages.

**Keywords:** loanwords; Spanish; Wayuunaiki; language contact; bilingualism

## 1. Introduction

Interest in loanwords—or lexical loans—has traditionally been driven by the goal of formulating typological links between languages around the world (Comrie 1988a, 1989, 2000; Greenberg 1957; Martin 1966). More recent work has expanded beyond a typological focus to include research on bilingualism and language contact (e.g., Alvanoudi 2018; Hoffer 1996; Muysken 2013; Poplack and Sankoff 1984; Poplack et al. 1988; Sasse 1985; Sousa and García 2020; Zenner et al. 2015). Within these fields of inquiry, loanwords are considered the primary forms of linguistic change resulting from contact situations (Calude et al. 2017; Grant 2015).

Recent studies have drawn attention to complex factors conditioning lexical loans, including lectal features (e.g., Swerts et al. 2021; Zenner et al. 2012), historical period (e.g., Chesley and Baayen 2010; Winter-Froemel et al. 2012), geographical proximity (e.g., Franco et al. 2019; Sousa and García 2020), and semantic features (e.g., Calude et al. 2017; Macalister 2008). Despite a growing understanding of these factors, the comprehension of loanwords in contact contexts is still limited, mainly due to the reduced set of languages studied and the exclusive focus of some studies on documenting loanwords without offering more profound accounts of the factors mediating their occurrence.

This study aims to contribute to the body of research on lexical borrowing in contact situations by identifying and analyzing the main loanwords from Spanish present in Wayuunaiki, an indigenous language spoken in the Guajira Peninsula in northern Colombia and northwestern Venezuela. Our analysis considers three factors related to loanwords:

their grammatical word class, their classification according to semantic domains, and the motivations behind their adoption. This analysis contributes to comprehending the mechanisms conditioning the uptake and assimilation of loanwords between typologically distant languages. More broadly, our research contributes to the growing body of literature on Wayuunaiki–Spanish language contact and bilingualism (e.g., Atencio 2014; Etxebarria Arostegui 2012; Mejía Rodríguez 2011; Méndez Rivera and Mancipe 2014; Méndez-Rivera 2020; Oquendo 2014; Pérez van Leenden 2003a, 2003b; Pimiento Prieto 2008).

The questions addressed in this paper are as follows:

(1)   Does Wayuunaiki allow a high or low degree of loanword incorporation?
(2)   Which semantic domains are more susceptible to loanword incorporation in Wayuunaiki–Spanish contact?
(3)   Does Wayuunaiki allow loans of word classes other than nouns (the dominant loanword type)? And to what extent?
(4)   What are the motivations behind the adoption of loanwords from Spanish?

This paper contains seven sections. In Section 2, we present a review of the relevant literature on loanwords. In Section 3, we provide information about the typological characterization of Wayuunaiki and Spanish and an overview of their contact dynamics. Section 4 describes the methodological procedures employed in this study. In Section 5, we present the findings. We continue in Section 6 with a discussion of the theoretical implications. Finally, in the Conclusion (Section 7) we address the research questions, summarize the implications of the findings, and suggest further lines of research.

## 2. Loanwords and Language Contact

Since contact between human groups is inherent to the human species, language contact is the rule, not the exception (Nicolaï 2019, p. 280). Prolonged contact may result in "new forms of language use" which contrast with those of monolingual communities (Romaine 2004, p. 49). From a geographical standpoint, contact may be indirect or direct. Indirect contact occurs when the recipient language borrows words from a donor language used in a different geographical space. This is the case with English and other European languages, whose contact mainly owes to the influence of television, radio, and the Internet (Hickey 2010). Indirect contact does not depend on a state of bilingualism and is present in the written language used by educated bilingual speakers (Sala 2013, p. 188). Due to globalization, indirect contact may be attested all around the world. Although borrowings related to cultural aspects may be found in indirect contact situations, effects on the grammar of the recipient language are not generally expected (Hickey 2010, p. 7).

Direct contact refers to the coexistence of two or more languages within a shared space. This pervasive form of linguistic contact generally involves a mixture of populations that may cohabitate during varying periods (Sala 2013). Unlike indirect contact, direct contact yields not only lexical loans but also structural transfer (Hickey 2010). Our work focuses on the direct contact between Wayuunaiki and Spanish in the Guajira Peninsula, located between Colombia and Venezuela.

In this work, we have adopted Hoffer's (2005, p. 53) definition of "loanword" as the incorporation by a recipient language of an item or idea with its corresponding word in the source language. Incorporating loanwords is a universal process to the extent that, very likely, no language is devoid of borrowings (Tadmor 2009). The universal nature of this phenomenon makes the study of borrowing a crucial component of historical–comparative linguistics (Haspelmath 2009). One of the issues that has raised interest is the existence of properties which make a language more susceptible to adopting loanwords. Tadmor (2009) considers this a complex question, with counterexamples defying simple answers and language-specific properties providing better-fitting explanations. In his 2009 study, for example, he found that the ten languages with the highest proportion of loanwords (Selice Romani, Tarifit, Gurindji, Romanian, English, Samaraccan, Ceq Wong, Japanese, Indonesian, and Bezhta) possess well-differentiated typological and sociolinguistic features.

To address this issue, Hoffer (1996) proposes the terms adaptability scale and receptivity scale. The adaptability scale refers to the ability of the phonological system of a language to adapt itself to the adoption of loanwords from various typologically distinct languages. Loanword adoption would thus be more complex when the recipient language has a more reduced consonant and vocalic inventory and/or a simpler syllabic structure and/or an overall highly differentiated intonational system (p. 61). Hawaiian provides an example of these properties since its vocal inventory of only five vocalic sounds and its simple consonant + vowel-type syllabic structure impairs the adoption of loanwords from languages with contrasting phonological and syllabic structure properties, like English.

The receptivity scale refers to the degree of acceptance of or resistance to loanwords (Hoffer 1996, p. 61). There are two aspects of receptivity: (1) the proportion of loanwords accepted by a language throughout its history and (2) the level of official resistance against loanwords posed by governmental bodies (p. 61). Regarding the acceptance aspect, Hoffer (ibid.) locates Spanish and English at the top of the scale given the large number of loanwords accepted by them throughout history. Chinese, in contrast, is located at the bottom of the scale given its reduced proportion of loanwords. From the resistance standpoint, Hoffer (ibid.) places English at the bottom of the scale due to the lack of official regulation against the entry of lexemes from other languages, in contrast to Chinese, where the official rejection of loanwords has been significant.

Besides recognizing the intrinsic linguistic features of recipient and donor languages, loanword theories attribute an important role to bilingualism in contact situations. Matras (2009, p. 312) points to the directionality of bilingualism as a major social factor influencing the spread of linguistic innovations. He argues (p. 312) that unidirectional bilingualism, where the recipient language is mostly employed in personal informal settings, drives users to adopt more structures from the dominant donor language because the bilingual mode is more frequently used in different forms of interaction[1]. This type of bilingualism is found in most contexts under study and reflects the accelerated process of language change and the disappearance of minority languages—a great majority of which are indigenous languages.

Aside from typological similarities and differences and the type of bilingualism found in specific cases, geographical closeness and proximity between dialects/languages may play a relevant role in the lexical transfer. One example is the study of loanwords in Galician in contact with Spanish and Leonese. Through dialectometric methods, Sousa and García (2020) used loanwords and dialectal maps from the Iberian Peninsula to compare different dialects of Galician. Their study showed two main patterns. On the one hand, the western dialects of Galician are distinguished by their homogeneity, proximity, population density, and low levels of loanwords from Spanish and Leones. On the other hand, eastern Galician dialects in stronger contact with Spanish and Leones showed higher numbers of loanwords from those languages and more autonomy in loanword incorporation.

Social norms within the community also constitute a major driving force in the acceptance or rejection of loanwords (Matras 2009; Poplack et al. 1988). Community social norms often cohere with the receptivity features proposed by Hoffer (1996), in that individual users may be willing to adopt a higher or lower number of loanwords depending on the perceived social status and prestige of the donor language. In the case of the Wayuu community in the middle zones of the Guajira Peninsula, a unidirectional bilingual situation is observed, wherein Wayuunaiki is used in personal and family contexts and Spanish increasingly gains ground in other social realms (Mejía Rodríguez 2011). Wayuunaiki may thus be hypothesized to be highly receptive to loanwords given the sociolinguistic dynamics of Wayuu communities. Before investigating this hypothesis, a brief consideration of the typological features of Wayuunaiki and Spanish is in order.

## 3. Main Features of the Recipient and Donor Languages

Wayuunaiki (also known as Guajiro or Wayuu) is the most widely spoken Arawak language nowadays. Wayuunaiki speakers inhabit the Guajira Peninsula, located between

the Colombian Caribbean region and northwestern Venezuela. Its location far up north makes Wayuunaiki one of the most geographically distant Arawak languages, since others within this linguistic family are mostly spoken in the Amazon region and in countries such as Brazil, Peru, Bolivia, Surinam, and Venezuela (Aikhenvald 1999). Around 416,000 people are currently estimated to use it as a native language in Colombia and Venezuela (Ethnologue n.d.b).

The Lower and Middle Guajira Peninsula contains the highest proportion of Wayuunaiki–Spanish bilingual speakers, whereas Upper Guajira is the place where most monolingual Wayuunaiki speakers can still be found. Despite Wayuunaiki being the most widespread Arawak language, recent studies have found its native speaker population to be in decline, with the number of monolingual Spanish speakers in the region increasing (Etxebarria Arostegui 2012; Mejía Rodríguez 2011; Pérez van Leenden 2000a, 2000b, 2003a, 2003b). This increasing imbalance poses a threat to the linguistic vitality of Wayuunaiki.

Like other languages in the Arawak family, Wayuunaiki is an agglutinating VSO word-order language (Pimiento Prieto and Van-Leeden 1997). Wayuunaiki allows the occurrence of null subjects and does not mark the genders for the first and second persons (only for the third) in constructions with free subject pronouns (Viloria Rodríguez et al. 2022). Regarding its phonological inventory, Wayuunaiki makes use of 12 vowel sounds and 14 consonant sounds (see Mansen 1967), as shown in Table 1.

**Table 1.** Wayuunaiki vowel and consonant inventory.

| | Vowel Phonemes | | | | | |
|---|---|---|---|---|---|---|
| | Front | | Central | | Back | |
| | Short | Long | Short | Long | Short | Long |
| High | i | i: | ü | ü: | u | u: |
| Middle | e | e: | | | o | o: |
| Low | | | a | a: | | |
| | Consonant Phonemes | | | | | |
| | Bilabial | Alveolar | Post-Alveolar | Palatal | Velar | Glottal |
| Plosive | p | t | | | k | *P* |
| Nasal | m | n | | | | |
| Fricative | | s | | *S* | | h |
| Affricate | | | | | | |
| Lateral | | l | | | | |
| Vibrant | | r | | | | |
| Approximant | w | | | j | | |

Other distinctive features of Wayuunaiki include the use of alienable and inalienable possessive markers (Ramírez González 2021), analytical and synthetic conjugations (Olza Zubiri and Jusayú 1978), two grammatical genders (with feminine as unmarked) (Regúnaga 2005), absolute and relative nouns (Álvarez 2017), and a clear distinction between transitive and intransitive verbs. This latter distinction is considered to be gradual due to the existence of different mechanisms to reduce or increase verbal valence.

On the other hand, Spanish is spoken in more than 21 countries. This Romance language has a strong linguistic vitality due to the current increase in Spanish L1 and L2 speakers around the world. According to some estimates, there are approximately 471.397.370 speakers of Spanish as a first language and 71.497.140 speakers of it as a second language (Ethnologue n.d.a). With some variation across dialects, Spanish has an inventory of 5 vowel sounds and 18 consonant sounds (Salcedo 2010). Table 2 shows the distribution of sounds according to their characteristics:

**Table 2.** Spanish vowel and consonant inventory.

| Vowel Phonemes | | | | | | |
|---|---|---|---|---|---|---|
| | Front | | Central | | | Back |
| High | i | | | | | U |
| Middle | e | | | | | o |
| Low | | | a | | | |
| **Consonant Phonemes** | | | | | | |
| | Bilabial | Labiodental | Alveolar | Post-Alveolar | Palatal | Velar | Glottal |
| Plosive | p b | | t d | | | k g | |
| Nasal | M | | n | | ɲ | | |
| Trill | | | r | | | | |
| Tap or Flap | | | ɾ | | | | |
| Fricative | | f | s | | | x | |
| Affricate | | | | ʧ | | | |
| Glides | W | | | | j | | |
| Liquid | | | l | | | | |

Other distinctive features in Spanish include a free word order with an SVO tendency and fusional inflection (Almela Pérez 2015; Mijangos and de León 2017) and a rich verb agreement system (Comrie 1988b). Like Wayuunaiki, Spanish subject pronouns do not mark gender for first and second singular persons, but they do for third singular and plural persons (Viloria Rodríguez et al. 2022).

Both languages coexist as official languages in the Guajira Peninsula (northern Colombia and northwestern Venezuela). Unlike in other regions of Colombia and Latin America, Spanish was slow in becoming established in this region due to the remote location of the peninsula and the scarcity of natural resources for survival within its mostly desert environment (see Landaburu 2005). These conditions favored the preservation of the indigenous language up to the present day. One of the consequences of the increasing contact between both languages is the decrease in Wayuunaiki L1 speakers in the region. Despite monolingual Wayuunaiki speakers still being relatively numerous in Upper Guajira, Spanish continues to gain ground in the lower and middle peninsula regions. This change has resulted in Spanish now being spoken in many homes where Wayuunaiki was traditionally spoken (Mejía Rodríguez 2011).

Recent studies on Wayuunaiki–Spanish contact have reported a lack of gender and number agreement in Spanish L2 speakers (Ramírez 2012) and reverse masculinization and feminization of nouns in Spanish, as in *serrucha* from the Spanish masculine noun *serrucho* 'saw', *sementa* from the Spanish masculine noun *cemento* 'cement', and *kolegia* from the masculine Spanish noun *colegio* 'school' (Oquendo 2014, p. 150). These traits—numeral pluralization and gender switch, as shown in examples (1) and (2)—represent a common feature of Wayuu bilinguals, especially in Lower and Middle Guajira:

(1)  Mi          mamá        tuvo          nueves        hijos.
     My          mom         have-PAST     nine-PL       children-M
     'My mom had nine children'.

(2)  Yo    pienso      otro,        otro          nuevo,      distinto       [una   nueva  mujer].
     1SG   think-1SG   another-M    another-M     new-M       different-M    [a     new    woman]
     'I think of another, another new, different [a new wom-an]'.

(Examples taken from Ramírez 2012, pp. 674–81; translations into English and glosses are added).

In example (1), the speaker pluralizes the numerical word *nueve* 'nine'. This trait is unusual in Spanish, since the original word implies a plural meaning. In the second example, when referring to a woman, the speaker omits the agreement in *otro* 'another' (instead of *otra*), *nuevo* 'new' (instead of *nueva*), and *distinto* 'different' (instead of *distinta*), which are typically marked with *-a* along with determiners as a signal of female gender agreement. In addition, the speaker omitted the preposition *en* 'in' which is generally used with *pensar* 'to think of'. In this example, it would be enough to say *Yo pienso en otra mujer*, which implies the idea that she is someone different and new to the speaker.

Another phenomenon includes the fusion of grammatical elements with syntactic elements of both languages in code-switching, as shown in examples (3) and (4). The words in Spanish are in bold to differentiate them from those in Wayuunaiki:

| (3) | Porque | tampoco | kasa-in | ta-nain. |
|-----|--------|---------|---------|----------|
|     | Because | neither | that-SUB | 1SG-ADHESS |
|     | 'Because I don't know what happens to me either'. | | | |

| (4) | Aishajaa, | ¿qué | problemas | tienes? |
|-----|-----------|------|-----------|---------|
|     | Gee | what | problem-PL | have-2SG |
|     | 'Gee, what are your problems?' | | | |

The last examples illustrate code-switching by bilingual speakers of Spanish and Wayuunaiki. Example (3) corresponds to an alternation in a sentence that starts in Spanish and continues in Wayuunaiki. Example 4 illustrates what Muysken (2013, p. 713) calls *back-flagging*, that is, the use of discourse markers from the speaker's native language as an index of ethnic identity. Although examples of code-switching are frequent in bilingual Spanish–Wayuunaiki speakers, to date, this topic lacks scrutiny.

Another common phenomenon found in the Spanish spoken by the Wayuu is the devoicing of velars in onset position (i.e., SP *mango* > *mankü* [maŋkü] 'mango', SP *gaseosa* > *kaseoosa* [kaseosa] 'soda'). This tendency is recurrent in words borrowed from Spanish into Wayuunaiki, in consonance with the phonological inventory of the borrower language. The same phenomenon is found in some variants of spoken Spanish.

As we have seen in this section, Wayuunaiki and Spanish are typologically distant languages. They show phonological and grammatical differences that may represent a challenge for the adoption of words from one to the other. However, this challenge is unlikely to be a decisive factor for the adoption of lexical and grammatical features since social factors can prevent or facilitate lexical borrowing. This study examines the degree to which Wayuunaiki incorporates Spanish loanwords, the semantic domains more susceptible to borrowing, the main types of borrowed words, and the possible motivations behind the adoptions. In the following section, we describe the type of data used in this study and how they were processed.

## 4. Materials and Methods

Our study was based on the integration of documentary source analysis and fieldwork. Documentary evidence was obtained via the analysis of lexical entries from four Wayuunaiki–Spanish dictionaries: *Diccionario básico ilustrado Wayuunaiki-Español/Español-Wayuunaiki* (Captain and Captain 2005), *Diccionario Wayuunaiki-Español/Español-Wayuunaiki* (Ramírez González 2008), *Diccionario ilustrado Wayuunaiki-Español* (Romero 2012), and *Diccionario Wayuu–Inglés–Español* (Captain and Captain 2019). The field data correspond to data collected in the Middle and Upper Guajira Peninsula in 2022. These data were employed as a source of evidence of lexical items used in more colloquial everyday contexts often not represented in dictionaries. The interviews were conducted by a Wayuunaiki native speaker from the community, and each lasted approximately 40 min. Fifty-five people gave their consent and voluntarily participated in the interviews and other elicitation activities that were part of a bigger project focused on the description of the linguistic contact between Wayuunaiki and Spanish. For this specific study, a random subsample of 6 bilingual speakers (3 male and 3 female participants whose ages ranged from 18 to 71 years)



and 4 monolingual speakers (2 males and 2 female participants whose ages ranged from 50 to 71 years) were selected to analyze their oral productions and identify other borrowed words not found in dictionaries.

The analysis only included lexical entries that appeared in more than one dictionary or whose usage was declared valid by Wayuunaiki L1 speakers. The lack of common typological closeness between Spanish and Wayuunaiki was found to facilitate loanword identification (Haspelmath 2009; Haspelmath and Tadmor 2009). Another facilitating feature was Captain and Captain's (2019) explicitly signaling loanwords from Spanish in their dictionary. In identifying loanwords, we made sure to include only those lexical items that were unanalyzable in the recipient language. Following Haspelmath (2009, p. 37), analyzable items cannot be considered loanwords because they are "created within the recipient language" and they generally lose their original structure when adopted within it.

The loanword status of lexical entries appearing only in field data was validated with the aid of a Wayuunaiki expert, who, in turn, consulted two Wayuunaiki L1 speakers in ambiguous cases. This double validation allowed us to avoid the inclusion of nonce borrowings, that is, single-use items whose usage is not generally recognized by Wayuunaiki L1 users or dictionaries. These nonce borrowings may be a consequence of code-switching and not of a systematic borrowing process (Haspelmath 2009; Poplack 2018).

After extracting lexical entries and validating their loanword status, we classified them according to their grammatical classes (noun, verb, adjective, etc.) and semantic domains. The semantic classification was based on Haspelmath and Tadmor's (2009) model. Among the domains, we found religion and belief, clothing and grooming, the house, law, social and political relations, agriculture and vegetation, food and drink, warfare and hunting, possession, animals, cognition, basic actions and technology, time, speech and language, quantity, emotions and values, the physical world, motion, kinship, the body, spatial relations, and sense perception. We also checked the previously identified lexical entries against the ones proposed in the Swadesh list (Swadesh 1955) and in the Leipzig–Jakarta basic vocabulary list (Tadmor 2009) to assess correspondence with general loanword trends.

## 5. Results

A total of 536 lexical items were classified as Spanish loanwords based on the analysis of dictionaries and field data. Interestingly, none of these borrowed items appear in the Swadesh list (Swadesh 1955) or in the Leipzig–Jakarta list (Tadmor 2009). Since both lists include items that are less prone to borrowing, strong similarities between languages based on the elements of these lists may suggest greater possibilities of linguistic filiation.

In Table 3, we can see the number of words per dictionary, along with the percentages of loanwords adopted from Spanish into Wayuunaiki, ranging between 3.7% and 11.2%. The highest percentage of borrowed words corresponds to the second dictionary, which was created by a member of the Wayuu community.

**Table 3.** Number of loanwords per dictionary.

| Name of Dictionary | Number of Words | Numbers and Percentages of Loanwords per Dictionary |
|---|---|---|
| Diccionario básico ilustrado Wayuunaiki-Español/Español-Wayuunaiki (Captain and Captain 2005) | Wayuu–Spanish: 1361 | 77 (5.6%) |
| Diccionario Wayuunaiki-Español/Español-Wayuunaiki (Ramírez González 2008) | Wayuu–Spanish: 1347 Spanish–Wayuu: 1129 | 151 (11.2%) |
| Diccionario ilustrado Wayuunaiki-Español (Romero 2012) | Wayuu–Spanish: 1380 | 52 (3.7%) |
| Diccionario Wayuu–Inglés–Español (Captain and Captain 2019) | Wayuu–Spanish: 3092 Spanish–Wayuu: 2485 | 154 (4.9%) |

As can be seen in Table 3, all the dictionaries contain a moderate number of words, ranging between 1347 and 3092. These limited numbers suggest that many Spanish loanwords fail to be documented in dictionaries. In fact, only 266 loanwords out of the total of 536 loanwords identified in this study (49.62%) appear in these sources. The remaining 270 loanwords (50.37%) were identified in semi-spontaneous conversations and validated through inquiry with three L1 Wayuunaiki speakers from the community, as explained in the Materials and Methods section.

Regarding grammatical word classes, Figure 1 shows that Spanish loanwords in Wayuunaiki fall mainly within the categories of nouns, verbs, and adjectives. This finding is in consonance with general trends observed for loanword grammatical classes (see Matras 2009):

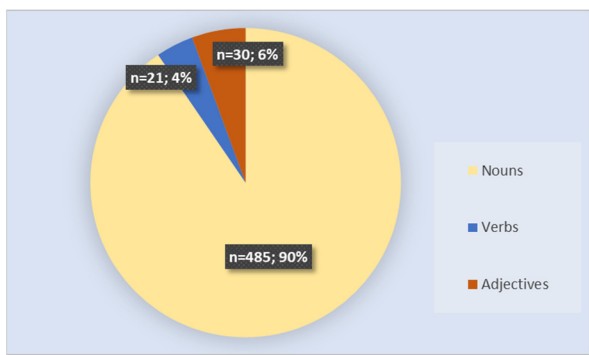

**Figure 1.** Distribution of loanwords in word classes.

Nouns constitute the most frequent grammatical category (n = 485, 90%), followed by adjectives (n = 30, 6%) and verbs (n = 21, 4%). Most of the adjective loanwords (n = 25 out of 30) can also function as nouns, as in Spanish (e.g., *püreesaa* from the Spanish *preso*, meaning 'prisoner' as a noun or 'in prison' as an adjective). The rest of the grammatical word classes, including adverbs, auxiliaries, and prepositions, were not found in the dictionaries or field data.

Below, we provide some examples of loanwords in their respective sentences. All of them were taken and adapted from the electronic dictionary *Diccionario Wayuu–Inglés–Español* (Captain and Captain 2019). Examples 5 and 6 illustrate the use of two loanwords that behave like verbs; example 7 illustrates the incorporation of adjectives; and examples 8 and 9, the incorporation of nouns.

(5) Nü-woutisaaj-ü-in apünüin-shii wayuu, woutiisa kalaka nia.
3SG.M-baptize-PST-SUB three-PL people, baptize later 3SG.M
'He baptized three people, later he was baptized'.

(6) Kasaat kalaka nia sü-maa.
married later 3SG.M 3SG.F-SOC
'He later married her'.

(7) Ee-shi wane wayuu ka-nülia-shi Barrabás,
there is-M a person ATRIB-name-M Barrabás,
püreesa-shi nia na-maa na nü-püshi-ka-na.
imprisoned-M 3SG.M 3PL-SOC DEF.PL 3SG.M-relative-DEF-PL
'There was a man named Barrabas, he was imprisoned with his fellows'.

(8) Awatatuu-sü tü ta-kachuucha-in-ka-t sü-tüma tü jouktai-ka-t.
blow off-F DEF 1SG-cap-POS-DEF-F 3SG-through DEF wind-DEF-F
'My cap blows off in the wind'.

(9) Atut-sü pi-yaweera-se.
worn out-F 1SG-keychain-POS
'Your keychain is worn out (falling to pieces)'.

Although addressing phonological and morphosyntactic adaptations is beyond the scope of this article and deserves closer scrutiny in future research, we can see that the previous examples of loanwords are adapted to the morphology and syntax of Wayuunaiki, especially verbs (as described in Wichmann and Wohlgemuth 2008), but also some nouns. These items seem to be fully adapted to the grammatical and phonological system of Wayuunaiki, to the extent that older speakers anecdotally claimed to be unaware of their foreign origin.

Regarding the semantic fields receiving most lexical borrowings, Figure 2 presents the results based on Tadmor's (2009) classification framework. The graph shows that most Spanish loanwords in Wayuunaiki fall within the category of *basic actions and technology* (213 out of 536; e.g., *sirii* from SP *CD* 'CD', *piila* from SP *pila* 'battery'). The rest fall, in descending order, within the domains of *food and drink* (n = 62, 11.5%; e.g., *aaju* from SP *ajo* 'garlic', *sewoya* from SP *cebolla* 'onion'), *social and political relations* (n = 52, 9.7%; e.g., *atkattia* from SP *alcaldía* 'town hall', *kantirato* from SP *candidato* 'candidate'), *clothing and grooming* (n = 46, 8.5%; e.g., *kamiisa* from SP *camisa* 'shirt', *korwaata* from SP *corbata* 'tie'), *religion and belief* (n = 26, 4.8%; e.g., *katooliko* from SP *católico* 'catholic', *miisa* from SP *misa* 'mass'), *the house* (n = 26, 4.8%; e.g., *kosinapia* from SP *cocina* 'kitchen', *wayo* from SP *baño* 'bathroom'), *time* (n = 25, 4.6%; e.g., *oora* from SP *hora* 'hour', *semaana* from SP *semana* 'week'), *the physical world* (n = 22, 4.1%; e.g., *lamuuna* from SP *laguna* 'lake', *montaya* from SP *montaña* 'mountain'), *animals* (n = 19, 3.5%; e.g., *kochiina* from SP *cochino/cerdo* 'pig', *pawo* from SP *pavo* 'turkey'), and *warfare and hunting* (n = 13, 2.4%; e.g., *pusiit* from SP *fusil* 'rifle', *surulaalü* from SP *soldado* 'soldier'). The other categories contain fewer than 10 items. No loanwords were found within the domain of *sense perception.*

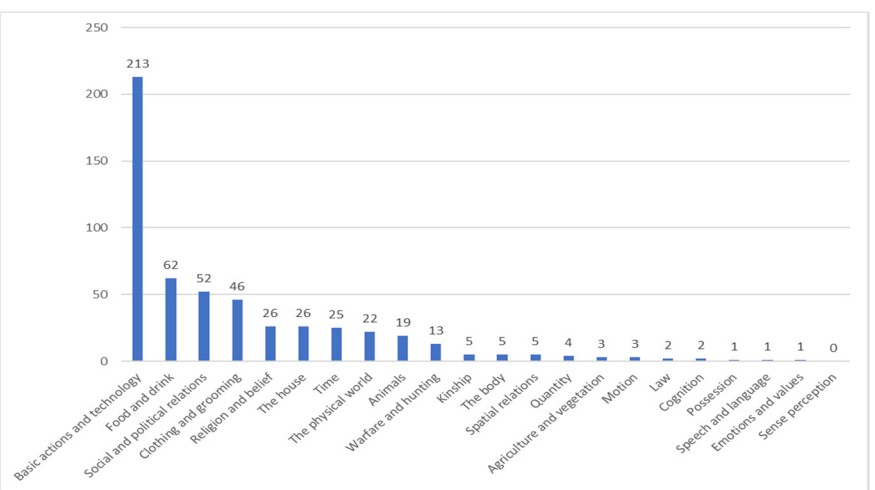

**Figure 2.** Loanwords per semantic domain based on Tadmor (2009).

The fact that most loanwords belong to the domains of *basic actions and technology*, *food and drink*, *social and political relations,* and *clothing and grooming* is unsurprising when considering that the Wayuu people have historically resisted cultural influences perceived as colonizing. The Wayuu people's acculturation process has been described as slow but irreversible (Hostein 2010, p. 5). The borrowing of words from these domains reflects their recent openness to the outside world, probably driven by commercial exchange with surrounding Spanish-speaking communities and tourism.

The borrowing of words from the domain of social and political relations can be understood as part of the Wayuu community members' gradual encroachment into the political arena. Participation in political and social activities has created the need to use terms originally missing in the Wayuu culture, whose political organization considerably differs from Western models in the sense that it is decentralized and composed of a confederation of dispersed clans (Sabogal 2018, p. 4).

The same acculturation dynamics apply to the domain of *religion and belief*. Although the Wayuu people have local religious beliefs and practices, many members of their community have converted to Protestantism, Catholicism, and other religions circulating in their territory (Pérez 2004). This religious assimilation is evident in the construction of Protestant and Catholic temples in the communities and in Wayuus' increasing attendance at Christian worship places. The church currently constitutes another major context for Spanish usage (Mejía Rodríguez 2011, pp. 80–81) that promotes the borrowing of terms that were not originally part of Wayuu religious traditions. In fieldwork interviews, some Wayuu participants claimed to be unaware of their own tribal religion, which could facilitate further lexical borrowing in the religious domain.

The low number of words borrowed within the domains of *kinship* (e.g., *kumaare* from SP *comadre* 'godmother of somebody´s child', *mitia* from SP *mi tía* 'my aunt'), *the body* (e.g., *müreena* from SP *moreno* 'dark-skinned', *ripunto* from SP *difunto* 'deceased body'), *spatial relations* (e.g., *kayejon* from SP *callejón* 'alley', *puetto'u* from SP *puerto* 'port'), *cognition* (e.g., *enseyansa*, from SP *enseñanza* 'learning/lesson', *epensaajaa* from SP *pensar* 'think'), *possession* (e.g., *poropietario* from SP *propietario* 'owner'), and *emotions and values* (e.g., *a'yalajaa* from SP *llorar* 'cry') shows that Wayuunaiki is still relevant in the family and personal contexts. This finding is in consonance with previous sociolinguistic descriptions by Etxebarria Arostegui (2012) and Mejía Rodríguez (2011). Both authors stated that Wayuunaiki is still preferred by many members of the community in family and intimate contexts. Regarding *agriculture and vegetation* (e.g., *patsera* from SP *parcela* 'plot to plant', *mariwana* taken from SP *mariguana* 'marijuana')*,* low rates may be explained by the arid nature of much of the Wayuu territory and the consequent irrelevance of agriculture as an economic activity in the region. Although some of the plants consumed in the region have Spanish names, during the interviews, we found that the Wayuu prefer vernacular terms. As for *quantity measurement* (e.g., *kila* from SP *kilo* 'kilo', *turoosa* from SP *trozo* 'slice/piece of something'), our field data show that the Wayuu continue to use the well-developed lexical repertoire offered by their L1 in this domain. Although the word *cero* 'zero' was not found in any of the sources consulted and is not part of the Wayuunaiki numbering system, it can also be considered a loanword from Spanish. This word is only used in necessary cases, for instance, in the use of passwords, codes, or mathematical operations that require it.

Since the category of *basic actions and technology* contains a large proportion of loanwords, we found it useful to subdivide it into further domains, as shown in Figure 3. Most loanwords fall within the sub-category of *tools, materials, and utensils* (n = 79, 36%; e.g., *alaampira* from SP *alambre/cable* 'wire/cable', *woteya* from SP *botella* 'bottle'), followed by *science and technology* (n = 38, 7%; e.g., *kaamara* from SP *cámara* 'camera', *wompiiya* from SP *bombilla(o)* 'electric light bulb'), *institutions* (n = 24, 4.4%; e.g., *misoora* from SP *emisora* 'radio station', *püneraria* from SP *funeraria* 'funeral home'), *actions* (n = 14, 2.6%; e.g., *epesajaa* from SP *pesar* 'weigh', *emiriijaa* from SP *medir* 'measure'), *culture* (n = 14, 2.6%; e.g., *katnawaar* from SP *carnaval* 'carnival', *muusika* from SP *música* 'music'), *education* (n = 14, 2.6%; e.g., *pakultat* from SP *facultad* 'department/school'; *tareeya* from SP *tarea* 'homework'), *transport* (n = 12, 2.2%; e.g., *mooto* from SP *moto* 'motorcycle', *wotkeeta* from SP *volqueta* 'dump truck'), *medicine and health* (n = 10, 1.8%; e.g., *wakuuna* from SP *vacuna* 'vaccine', *wenta* from SP *venda* 'bandage'), and *sports* (n = 8, 1.4%; e.g., *estario* from SP *estadio* 'stadium', *putwot* from SP *fútbol* 'soccer/football').

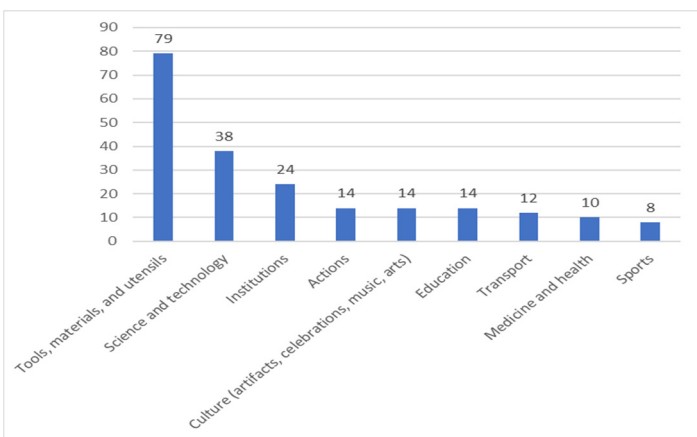

**Figure 3.** Loanword classification into sub-domains within *Basic actions and technology*.

A close inspection of the subdomains in Figure 3 shows that the borrowing from Spanish may be motivated by the absence of certain concepts in Wayuu culture. Some items correspond to the cultural adoption of new tools, materials, and medicines used in surrounding communities. The influence of the Colombian and Venezuelan governments in the Wayuu territory has prompted the adoption of educational, health, and administrative concepts previously unknown to the Wayuu. Contact between the Wayuu and other communities has also led to the adoption of celebrations and sports foreign to the ancestral culture. Nowadays, celebrations such as Christmas, Holy Week, New Year, and the Virgen del Carmen festivity, among others, are part of some Wayuu communities. The social uptake of these cultural influences is accompanied by the adoption of new concepts and their respective lexical borrowings.

One issue that attracted our attention is that some concepts are represented with two or even three forms. Some examples are *joojoro/pooporo* from SP *fósforo* 'match', *yawero/yaweerü* from Spanish *llavero* 'key ring', *owoutisaajaa/woutiisaa/awoutisaajaa* from Spanish *bautizar* 'baptize', and *maikki/maikkü/maiki* from Spanish *maíz* 'corn'. In fact, 61 borrowed words from the sample (11.3%) have more than one adapted form in Wayuunaiki. This pattern may be related to the phonology of both languages and awaits further research.

In addition to the existence of borrowed words with more than one form, there are 63 cases (11.7% of the total number of identified borrowed words) of two forms with two different origins, one from the receptive language and the other from the donor language, as in *jawon* from Spanish *jabón* 'soap' and *shupuuna* from Wayuunaiki; *kulira* from Wayuunaiki and *ansuelo* from Spanish *anzuelo* 'fish hook'; *saat* from Spanish *sal* 'salt' and *ichii* from Wayuunaiki; and *papeet* from Spanish *papel* 'paper' and *karalaukta* from Wayuunaiki. This finding is not surprising when it is taken into account that the coexistence of both forms may last for a long time, as previously attested in other language contact situations (Grant 2015, p. 434). This coexistence of Wayuunaiki and Spanish terms also suggests that lexical borrowing is not exclusively motivated by the importation of concepts missing in the recipient language. It could also be a matter of preferring borrowing as a facilitator of exchange with speakers of other languages.

The findings in this section describe the main phenomena involved in adopting Spanish loanwords into Wayuunaiki. The pattern shown by this indigenous language coincides with general trends regarding preference for the adoption of nouns and verbs. The dominant semantic domains observed include *tools, materials and utensils*, *food and drink*, *basic actions and technology*, and *social and political relations*. In the following section, we discuss the implications of these findings.

## 6. Discussion

The findings presented in Section 5 contribute to illuminating the research questions at hand. The first question is whether Wayuunaiki allows a large enough number of Spanish

loanwords for it to be considered highly receptive to this donor language. Our findings suggest that Wayuunaiki is moderately receptive to Spanish loanwords, as demonstrated by the number of words of Spanish origin (N = 536) found in the dictionaries and field data. Our data collection procedure followed different selection criteria to those in Tadmor (2009) because our search included items present in dictionaries and fieldwork, rather than being limited to a specific list. However, the number of items borrowed from Spanish demonstrates that Wayuunaiki is moderately receptive to borrowing words targeting technological, political, social, and medical needs at present. We may expect the number of loanwords to grow within these domains.

Although it is difficult and somewhat arbitrary to calculate the number of words in the Wayuu´s language due to the absence of a widely developed and available corpus or advanced lexicographic studies on the language, we could make some estimates if we consider the *Wayuunaiki-Spanish/Spanish-Wayuunaiki Dictionary* (Ramírez González 2008). We take this dictionary as a reference because it is the one that records the most lexical borrowings and was created by a member of the community who is characterized by a high degree of expertise in the indigenous language. It is possible that the other authors, looking at the language from an external and non-native point of view, could have avoided including some lexical borrowings either because of doubts about their real belonging to Wayuunaiki or to avoid contaminating the data with the dominant language. Based on the dictionary mentioned above, we found that, of the 1347 words, 11.2% correspond to lexical borrowings. If we take this percentage and compare it with the crosslinguistic data of Tadmor (2009, pp. 57–58), we could place Wayuunaiki in the category of "Average borrower", for which lexical borrowings represent approximately between 10% and 25% of the total sample. These estimates need to be confirmed in future studies.

Although some of the words borrowed and adapted from Spanish to Wayuunaiki were loanwords from English to Spanish (e.g., *penatti* from SP *penalti* 'penalty' or *turista* from SP *turista* 'tourist'), it is unlikely that they passed directly from English to Wayuunaiki because, on the one hand, the words reflect more Spanish than English phonology and, on the other hand, the contact between English and Wayuunaiki has not been extensive or prolonged. It is important to mention that these English-originated words, which have been borrowed by Spanish and later by Wayuunaiki, are widely used in the Spanish spoken in the region. In addition, no cases have been found of English words in Wayuunaiki that are not part of Spanish.

The results of this study also align with the general trend of basic vocabulary items being less prone to borrowing (Greenberg 1957, p. 39), which upholds the effectiveness of the lists proposed by Swadesh (1955) and the Leipzig–Jakarta essential vocabulary list (Tadmor 2009). The trends seen in loanword incorporation confirm that the needs concerning technology, politics, and social life lead Wayuunaiki speakers to adopt loanwords from Spanish. Spanish borrowings in Wayuunaiki are thus of the cultural type rather than of the basic type (in the sense of Haspelmath 2009). These needs become more relevant as the Wayuu become more involved in the political arena, adopt more technological elements, open up to new religions, and adopt more celebrations and customs from the dominant culture.

Technological and cultural uptake, however dominant, is unlikely to be the exclusive motivator of lexical borrowing in this contact setting. The coexistence of loanwords and vernacular terms referring to the same concepts, as identified in our study, suggests that Wayuunaiki speakers may strategically opt for loanwords to facilitate exchange with surrounding communities. The fact that 61 borrowed words from the sample (11.3%) have more than one adapted form in Wayuunaiki also raises the question of the relevance of phonological strategies and their competition in the adoption of borrowed words. This issue needs to be addressed in future investigations. In the same vein, future cross-linguistic comparisons may investigate whether the variation is typical of early stages of borrowing or may continue in other phases in which the borrowing process has decreased.

Regarding the question about the grammatical word classes receiving most loanwords, our study coincides with general trends by showing that Spanish loanwords in Wayuunaiki mostly correspond to nouns and, to a much lesser extent, adjectives and verbs. Prior studies (c.f., Matras 2009, pp. 167–68) have suggested that the preference for borrowing nouns may be related to their referential function, in contrast to the difficulties in morphosyntactically integrating verbs and adjectives into the recipient language. The analysis of morphosyntactic and phonological integration in Wayuunaiki awaits further research.

The fact that, in our data, many words (63 words; 11.7% of the sample) were identified as borrowings from Spanish, despite the existence of vernacular terms in the recipient language, is not a surprise when we consider that Spanish has been gaining in the Colombian department of La Guajira and along the western Venezuelan coast. As stated by Matras (2009, p. 312), in a unidirectional bilingual situation in which the recipient language is used in informal and intimate contexts, the bilingual mode drives the adoption of more structures from the donor language, since this mode is more frequently used in other types of interactions. Appealing to convenience by using loanwords in situations of widespread bilingualism constitutes an efficient strategy in communication (Haspelmath 2009, p. 47). Thus, the adoption of words from Spanish, even for terms that already exist in Wayuunaiki, suggests that the pressure of Spanish, as the dominant language around, is producing changes in the indigenous language.

An important consideration is that the Wayuu lack an official institution that regulates the inclusion or exclusion of loanwords. Purist attitudes towards other languages, i.e., "of cultural resistance to loanwords" (Haspelmath 2009, p. 47), have not been documented in this or prior studies. These social factors, combined with the cultural openness of the Wayuu towards the society around them and the prestige of Spanish in Colombia, create the right conditions for more loanwords to be incorporated into Wayuunaiki.

Our findings create the ground for refining hypotheses as to the mechanisms of lexical borrowing in similar contact situations. Accounts of word borrowability have taken into account phonological features from the donor and recipient languages, acceptance of or resistance to the adoption of borrowed words (Hoffer 1996), sociolinguistic factors (e.g., Poplack 2018), geographical proximity (e.g., Franco et al. 2019; Sousa and García 2020) (Haspelmath and Tadmor 2009; Hoffer 1996; Tadmor 2009), lectal features (e.g., Swerts et al. 2021; Zenner et al. 2012), and semantic features (e.g., Calude et al. 2017; Macalister 2008). Our study shows that other important sociohistorical factors include the openness of the recipient culture to external influences and the intensity of contact with dominant languages and cultures. These last two features may constitute some of the language-specific properties suggested by Tadmor (2009) that can contribute to providing better-fitting explanations. All these factors, taken together, may have a direct effect on lexical borrowing within specific semantic domains and not in others.

The implementation of large-scale initiatives, such as the Loanword Typology project (Haspelmath and Tadmor 2009), and small-scale projects like the one reported in this, and forthcoming studies can illuminate the dynamics of loanword incorporation in specific language contact situations. We believe that the time is ideal for new explanatory models to be proposed based on the growing documentation of lexical borrowing in language contact.

## 7. Conclusions

This study has shown that Wayuunaiki is moderately receptive to Spanish loanwords. The overall tendency is for Spanish loanwords to fill in terminological gaps referring to the semantic fields of basic actions and technology, food and drink, social and political relations, clothing and grooming, religion, and the physical world. All of them correspond to concepts present in the dominant Spanish-speaking culture that have been adopted into Wayuu culture owing to its increasing openness in recent decades. Although loanwords currently serve to fill these gaps, the intense contact between these languages could endow lexical borrowings with new meanings in the recipient language, owing to the cumulative

effect of prolonged contact ([Romaine 2004](#)). In line with general tendencies, Wayuunaiki mostly adopts nouns and, to a lesser extent, verbs and adjectives.

One limitation of the present study is the lack of a large Wayuunaiki corpus for studying loanword frequency. A corpus of spoken Wayuunaiki is especially called for given the strong oral tradition of Wayuu culture. Such a corpus would also enable the study of the morphosyntactic, phonological, and semantic adaptations in context. Although the present study has focused on speakers in the Colombian and Venezuelan territories, it would also be intriguing to establish comparisons in terms of frequency of use and lexical variation.

The scope of this article leaves out considerations of the phonological, morphosyntactic, and discursive integration of Spanish loanwords into Wayuunaiki. Detailed studies of these aspects could shed further light on the variation found in borrowed forms. Along the same lines, further research documenting loanword variation is necessary to comprehend variationist processes currently underway. This variation can also be compared with that in other language contact situations, which can be informative for linguistic theory.

Studies on language contacts are especially relevant in the Guajira region due to the co-existence of various indigenous languages, including Kogui, Wiwa (or Damana), Quechua Inga, and Kancuamo. Arabic is also spoken by a minority of Syrian–Lebanese immigrants in the northeastern town of Maicao. These languages could experience contact dynamics contributing to and drawing from the Wayuunaiki lexical repertoire. Although certain contact influences have been documented (e.g., [Ramírez 2012](#); [Oquendo 2014](#)), future studies are necessary to investigate bidirectional effects in contact between these languages, such as the adoption of Wayuunaiki loanwords into regional Spanish varieties. The expansion of Spanish in the region, indicated by the growing number of Spanish loanwords identified in Wayuunaiki, justifies more effective language preservation initiatives. Those initiatives must be supported by institutional actions that involve members of the Wayuu community as agents of change. These initiatives would also benefit from further efforts to document contact and bilingualism in the Colombian and Venezuelan Wayuu territories.

**Author Contributions:** Conceptualization, J.D.L.R.Y., R.R.G., L.V.R., W.V.J. and J.G.-L.; methodology, J.D.L.R.Y.; validation, J.D.L.R.Y., R.R.G., L.V.R., W.V.J. and J.G.-L.; formal analysis, J.D.L.R.Y., R.R.G., L.V.R., W.V.J. and J.G.-L.; investigation, J.D.L.R.Y., R.R.G., L.V.R., W.V.J. and J.G.-L.; resources, J.D.L.R.Y., R.R.G., L.V.R., W.V.J. and J.G.-L.; data curation, J.D.L.R.Y., R.R.G., L.V.R., W.V.J. and J.G.-L.; writing—original draft preparation, J.D.L.R.Y., R.R.G., L.V.R., W.V.J. and J.G.-L.; writing—review and editing, J.D.L.R.Y., R.R.G., L.V.R., W.V.J. and J.G.-L.; visualization, J.D.L.R.Y., R.R.G., L.V.R., W.V.J. and J.G.-L.; supervision, J.D.L.R.Y.; project administration, J.D.L.R.Y.; funding acquisition, J.D.L.R.Y. All authors have read and agreed to the published version of the manuscript.

**Funding:** This research was funded by the Dirección de Investigación, Desarrollo e Innovación Universidad del Norte, grant number 2021-008.

**Institutional Review Board Statement:** The study was conducted in accordance with the Declaration of Helsinki and approved by the Institutional Review Board (or Comité de Ética en investigación de la División Ciencias de la Salud) of Universidad del Norte (protocol code 264, approved on 28 April 2022) for studies involving humans.

**Informed Consent Statement:** Informed consent was obtained from all subjects involved in the study.

**Data Availability Statement:** Data available at: [https://osf.io/5cr9y/?view_only=b9b03539197341bc8c2ec9a14c124818](https://osf.io/5cr9y/?view_only=b9b03539197341bc8c2ec9a14c124818) (accessed on 30 March 2023).

**Acknowledgments:** We deeply appreciate the comments of our reviewers and the members of the Wayuu community who took part in this study. Without them, this study would not have been possible.

**Conflicts of Interest:** The authors declare no conflict of interest. The funders had no role in the design of the study; in the collection, analyses, or interpretation of data; in the writing of the manuscript; or in the decision to publish the results.

## Abbreviations

| | |
|---|---|
| 1PL | First plural person |
| 1SG | First singular person |
| 2PL | Second plural person |
| 2SG | Second singular person |
| 3PL | Third plural person |
| 3SG | Third singular person |
| ADHESS | Adhesive |
| ATRIB | Atributive |
| DEF | Definite |
| F | Feminine |
| M | Masculine |
| PL | Plural |
| POS | Possessive |
| PST | Past |
| SG | Singular |
| SUB | Subordinate |
| SOC | Sociative |

## Note

[1] Bilingual mode is a term coined by Grosjean (1985) to refer to the state of activation and language processing mechanisms of bilinguals at a specific time. The speaker may switch from a bilingual to a monolingual mode or vice versa depending on the characteristics of the interlocutor (see Grosjean 2013 for a more accurate description of mode change).

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
