# Peer review of "Lexical Borrowings from Spanish into Wayuunaiki: Contact, Classification, and Motivations"

_languages, doi:10.3390/languages8030169_

Round 1
Reviewer 1 Report
Review of “Lexical Borrowings from Spanish into Wayuunaiki: Contact, 2 Classification, and Motivations”
I really enjoyed reading this article, it is clear and organised and the structure makes it for a pleasurable read. The main contribution of the work is presenting evidence and discussion of Spanish borrowings into the Arawak language of Wayuunaiki, spoken in Latin America.
The findings are not surprising, for example, Wayuunaiki borrows extensively from Spanish (a dominant language in the region, with high prestige), and borrowings are typically nouns (as they are in other language contact scenarios), often denoting non-basic vocabulary and words for technology, time, religion and so on. However, the research is done diligently, explained well and it makes a worthy contribution to the existing literature on borrowing and language contact. For these reasons, I would very strongly recommend publishing this paper.
A few very minor issues noted below:
p. 7/18, line 275-276, the field work is not explained much, and is very vague (when was this conducted, how many informants were there, how older were they?) – a bit more information would be useful here
p. 10/18, line 364, ex. 7, there is a small typo here in the gloss line, “imprisioned” should be “imprisoned”
p. p. 11/18, line 424, I would suggest replacing “scenario” with “context”, might work better
p. 11/18, It might be useful to have some examples of borrowed words from Spanish here for each category. I realise that there are some examples given later on but it would be nice to have one for each category, especially for kinship, which according to Figure 2 has 5 tokens but interestingly, none are present in the Swadesh or Jakarta-Leipzig lists.
p. 12/18, line 490, I am a bit confused why there is “a preference for nouns and verbs”. To my mind, the preference is just for nouns (and if anything, it is adjectives, not verbs which make up the next highest group according to Figure 1).
Reviewer 2 Report
The study sheds light on an indigenous minority language and its contact with Spanish (the majority language). Sufficient context is provided, the use and review of the literature is solid, and the methodology used is sound (data are triangulated), making it a pleasurable read. Even though the manuscript does not include a strong hypothesis, and it is more exploratory in nature, it does provide an insightful examination of borrowings in Wayuunaiki.
I only have a couple of suggestions for the manuscript:
P.12, l. 498 “Wayuunaiki is highly receptive to Spanish loanwords”.
How is 536 items a high number? Can you provide a percentage, or a total number of (estimated) words in the language? Could you add whether English borrowings were found? If so, are they also borrowings in Spanish?
The other comment is actually a question: does this language have a word for “zero”? Is it a borrowing from Spanish?
